

# Secondary ozone peaks in the troposphere over the Himalayas

Narendra Ojha[1], Andrea Pozzer[1], Dimitris Akritidis[1,2], and Jos Lelieveld[1,3]

[1]Atmospheric Chemistry Department, Max Planck Institute for Chemistry, Mainz, Germany
[2]Department of Meteorology and Climatology, School of Geology, Aristotle University of Thessaloniki, Thessaloniki, Greece
[3]Energy, Environment and Water Research Center, The Cyprus Institute, Nicosia, Cyprus

*Correspondence to:* Narendra Ojha (narendra.ojha@mpic.de)

**Abstract.** Layers with strongly enhanced ozone concentrations in the middle-upper troposphere, re-
ferred to as Secondary Ozone Peaks (SOPs), have been observed in different regions of the world.
Here we use the global ECHAM5/MESSy atmospheric chemistry model (EMAC) to (i) investigate
the processes causing SOPs, (ii) explore both their frequency of occurrence and seasonality, and (iii)
assess their effects on the tropospheric ozone budget over the Himalayas. The vertical profiles of
potential vorticity (PV) and a stratospheric ozone tracer ($O_3s$) in EMAC simulations, in conjunc-
tion with the structure of SOPs, suggest that SOPs over the Himalayas are formed by Stratosphere-
to-Troposphere Transport (STT) of ozone. The spatial distribution of $O_3s$ further shows that such
effects are in general confined to the northern part of India. Model simulated ozone distributions
and backward air trajectories show that ozone rich air masses, associated with STT, originate as
far as northern Africa and the North Atlantic Ocean, the Middle-East, as well as nearby regions in
Afghanistan and Pakistan, and are rapidly (within 2–3 days) transported to the Himalayas. Analysis
of a 15-year (2000-2014) EMAC simulation shows that the frequency of SOPs is highest during the
pre-monsoon season (e.g. 11% of the time in May), while no intense SOP events are found during
the July-October period. The SOPs are estimated to enhance the Tropospheric Column Ozone (TCO)
over the central Himalayas by up to 26%.
**1 Introduction**
Tropospheric ozone is a short-lived climate forcer (Shindell et al., 2012) and an air pollutant with
adverse effects on human health and crop yields (Monks et al., 2015, and references therein). The
effects of tropospheric ozone on crop yields and human health occur near the surface, whereas its
radiative forcing is shown to be strongest in the middle-upper troposphere (e.g. Lacis et al., 1990;



Myhre et al., 2013; Monks et al., 2015). The processes controlling tropospheric ozone in the middle
and upper troposphere can be different from those near the surface. The photochemistry involving
non-methane volatile organic compounds (NMVOCs) and carbon monoxide, in the presence of ni-
trogen oxides (NOx) primarily controls ozone pollution in the planetary boundary layer. In contrast,
dynamics involving Stratosphere-Troposphere Exchange (STE) play a key role in the middle-upper
troposphere (e. g. Holton and Lelieveld, 1996; Lelieveld and Dentener, 2000; Neu et al., 2014; Ojha
et al., 2014; Monks et al., 2015). Therefore, to quantify the relative contributions of photochemi-
cal and dynamical processes to the ozone budget and assess the climatic impacts of anthropogenic
ozone, studies of the vertical distribution of ozone are essential.

Ozone observations have been conducted globally and locally using different instruments and

platforms as reviewed recently by Tanimoto et al. (2015). Balloon-borne observations employing
ozonesondes offer the advantage of measuring ozone across the tropopause. Analyses of ozonesonde
observations have provided valuable information on the variability, general features and trends in
ozone profiles (e.g. Logan, 1985, 1994). Secondary maxima in ozone profiles, called Secondary
Ozone Peaks (SOPs), are a unique phenomenon in which anomalously large ozone concentrations
are observed in confined layers in the middle-upper troposphere or lower stratosphere.

The occurrences of SOPs, underlying processes and their global distribution have been discussed

in a limited number of studies (Dobson, 1973; Reid and Vaughan, 1991; Varotsos et al., 1994), re-
viewed by Lemoine (2004). SOPs have been commonly observed in high latitudes, for example,
as laminated structures of ozone with the highest frequency of occurrence during the spring season
(Dobson, 1973). These laminated structures are primarily considered to be a winter-spring phe-
nomenon, with a peak altitude of occurrence near 14 km (Reid and Vaughan, 1991). Varotsos et al.
(1994) suggested that the north and northwest atmospheric circulations in the lower stratosphere play
a key role in the formation of SOPs observed over Athens, Greece. Overall, the occurrence of SOPs
is typically considered to be a northern hemispheric phenomenon, with no SOPs reported in the
tropics and the southern hemisphere (Lemoine, 2004). Trickl et al. (2011) showed the influences of
ozone import from the stratosphere and transport along the subtropical jet stream over Europe. Ac-
cording to the aforementioned studies, SOPs are mainly attributed to dynamical processes involving
STE, advection and Rossby wave breaking events.

Recent studies (Hwang et al., 2005, 2007; Park et al., 2012) focusing on the Korean region showed

that SOP events regularly occur over mid-latitudes. In contrast to earlier studies presenting SOPs
mostly in the lower stratosphere, several SOPs were observed in the upper troposphere over Korea.
Hwang et al. (2005) attributed these SOPs to the downward transport of ozone from the stratosphere
within a timescale of about one day (24 h), typical of the cross-tropopause exchange. Furthermore,
the frequency of occurrence, estimated from 9 years of ozonesonde observations, was found to have
strong seasonal variability over Korea with a broad winter-spring maxima and frequencies of occur-
rence up to 50–80 % (Hwang et al., 2005). Moreover, Hwang et al. (2005) reported an increase in



SOP occurrences over Korea, while the STT effects are anticipated to increase tropospheric ozone
in the future (Banerjee et al., 2016).
The studies pertaining to the influences of STT on the vertical profiles of ozone are relatively
sparse over the tropical Indian region. Mandal et al. (1998) analyzed observations from ozonesondes
and an MST Radar, and attributed the enhanced ozone mixing ratios in the upper troposphere to STT
through the indistinct tropopause over southern India. Fadnavis et al. (2010) combined satellite-borne
measurements (TES and MLS) with simulations performed by the MOZART model and showed sig-
nificant influences of STT over India in particular during winter and spring /pre-monsoon seasons.
Venkat Ratnam et al. (2016) used satellite observations to estimate the effect of STE associated
with tropical cyclones over the north Indian Ocean. Most of the studies based on in situ measure-
ments have however been confined over the southern part of India (e.g. Mandal et al., 1998; Sinha
et al., 2016) and the adjacent marine regions (Lal et al., 2013). Ganguly and Tzanis (2011) used
ozonesonde observations from three Indian stations operated by the Indian Meteorological Depart-
ment (IMD) and suggested that overall STT only plays a minor role into the budget of tropospheric
ozone over India. However, the influences of STT were found to increase with latitude /northward
over India (Ganguly and Tzanis, 2011).
Studies investigating the SOP structures and implications have been few over the tropical Indian
region, until very recently (Ojha et al., 2014; Das et al., 2016). The events over southern India were
found to be mainly associated with stratospheric intrusions during tropical cyclonic storms (Das
et al., 2016). In contrast, the SOP events observed over the central Himalayas in northern India ap-
pear similar to what is typically observed over the mid-high latitudes as mentioned earlier. Moreover,
SOPs were observed to be more frequent during spring, and were attributed to the combined effects
of STE and advection (Ojha et al., 2014). In the previous work using weekly ozonesonde measure-
ments (3–4 profiles per month), covering the period January 2011–December 2011 (Ojha et al.,
2014), only 6 SOP events were observed, being insufficient to calculate the frequency and seasonal-
ity of SOP occurrences. Additionally, model simulations are required to both trace the source regions
and quantify the effect of SOPs on the tropospheric ozone budget. Such investigation is of critical
importance over central Himalayas, as satellite-based studies show high pollution loading over the
northern India and the nearby Indo-Gangetic Plain (IGP) including the Tropospheric Column Ozone
(TCO) over South Asia (Fishman et al., 2003).
In the present study, the global atmospheric chemistry climate model EMAC (ECHAM5/MESSy
Atmospheric Chemistry) has been used to explore the processes causing the SOPs, investigate the
frequency and seasonality of their occurrence and finally assess their impact on the tropospheric
ozone budget over the central Himalayas.





## 2 Methodology

### 2.1 EMAC

The ECHAM5/MESSy Atmospheric Chemistry (EMAC) is a numerical system for the simulation of regional and global air quality and climate (Jöckel et al., 2010). In this work the model results from simulation RC1SD-base-10a of the ESCiMo project (Jöckel et al., 2016) are used. The general circulation model ECHAM5 version 5.3.02 (Roeckner et al., 2006) and the Modular Earth Submodel System (MESSy) version 2.51 (Jöckel et al., 2016) were used at T42L90MA-resolution, implying a spherical truncation of T42 (corresponding to a quadratic Gaussian grid of approx. 2.8 by 2.8 degrees in latitude and longitude) and 90 vertical hybrid pressure levels up to 0.01 hPa. The dynamics of the general circulation model were weakly nudged by Newtonian relaxation towards ERA-Interim reanalysis data (Dee et al., 2011). Gas-phase and particulate trace species calculated with the EMAC model have been extensively evaluated in previous studies (e.g. Pozzer et al., 2007, 2010, 2012). Simulation RC1SD-base-10a was selected between the ESCiMo simulations as suggested in Jöckel et al. (2016) ("For intercomparison with observations, we recommend to use the results of [...] RC1SD-base-10a."). Detailed information on the model set-up and comparison with observations can be found in Jöckel et al. (2016).

A tracer of stratospheric ozone, denoted as $O_3s$ in EMAC, has been used to quantify the effects of STT. $O_3s$ follows the transport and destruction processes of ozone in the troposphere but not its chemical formation (Roelofs and Lelieveld, 1997) and it is initialized to $O_3$ in the stratosphere.

### 2.2 Tropopause height and tropopause folds

The Lapse Rate Tropopause (LRT) height is calculated from EMAC output using the WMO definition as the altitude at which lapse rate decreases to a value of 2 ºC/km or less, provided that the average lapse rate between this level and all higher levels within the adjacent 2 km do not exceed 2 ºC/km.

Tropopause folds in EMAC simulations were identified with an algorithm developed by Sprenger et al. (2003), and improved by Ŝkerlak et al. (2014), using the three dimensional fields of potential vorticity, potential temperature and specific humidity. The vertical extent of the folds, as determined by the difference between the upper and middle tropopause crossings (see Fig. 1 in Tyrlis et al. (2014)) has been further used to identify shallow, medium and deep folds, as described and used elsewhere (Tyrlis et al., 2014; Ŝkerlak et al., 2015; Akritidis et al., 2016).

### 2.3 Observational dataset

The occurrence of SOPs was reported using ozonesonde observations from Nainital (79.45° E, 29.37° N, 1958 m asl), a high altitude station located in the central Himalayan region (Ojha et al., 2014). These data have been used to evaluate the capability of EMAC to reproduce SOPs over this





region. A typical event of SOP occurrence at Nainital observed on 10[th] March 2011 is shown in
Fig. 1a. The ozone mixing ratios in the middle-troposphere (10–11 km) are clearly observed to be
very high (150–250 nmol mol[-1]) forming an SOP. The location of Nainital station and the geograph-
ical topography of the northern Indian region are also shown in Fig. 1b.

Ozone profiles at Nainital were measured using Electrochemical Concentration Cell (ECC) ozoneson-

des. The method utilizes the titration of ozone in potassium iodide solution, which leads to produc-
tion of Iodine ($I_2$). The conversion of $I_2$ to $I^-$ in the cell leads to the flow of two electrons for each
ozone molecule entered. The measured cell current, flow rate of air along with sensor parameters,
e.g. the background current and pump temperature, are used to derive ozone mixing ratios (Ojha
et al., 2014). The precision and accuracy of ECC-ozonesondes are reported to be $\pm(3–5)\%$ and
$\pm(5–10)\%$ respectively, up to 30 km altitude (Smit et al., 2007).

Further details of the Nainital station and meteorology (Sarangi et al., 2014; Singh et al., 2016)

and balloon-borne measurements (Smit et al., 2007; Ojha et al., 2014; Naja et al., 2016) can be found
elsewhere.

## 2.4   Backward trajectories

We used the Hybrid Single Particle Lagrangian Integrated Trajectory (HYSPLIT) model (http://
ready.arl.noaa.gov/HYSPLIT.php) to investigate the source regions and the transport patterns caus-
ing SOPs over the central Himalayas. Backward trajectories have been simulated at 10, 11 and 12
km above sea level (asl), which are the typical altitudes where SOPs are observed in this study.
HYSPLIT trajectory simulations are driven by NCEP reanalysis meteorological fields and the model
vertical velocity option has been used for the vertical motions. More details of the backward trajec-
tory simulations using the HYSPLIT model (Draxler and Hess, 1997, 1998; Draxler et al., 2014) and
use of various datasets as meteorological inputs over the Indian region can be found elsewhere (e.g.
Ojha et al., 2012; Kumar et al., 2015).

## 3   Results and Discussion

### 3.1   Model Evaluation

Fig. 2 shows the comparison of EMAC simulated ozone profiles with ozonesonde measurements
over Nainital during six SOP events reported previously (Ojha et al., 2014). Model ozone fields
have been bilinearly interpolated to the observation site and model output closer to the time of
observation is weighted higher. As the vertical resolution of EMAC simulations is about 500–600
m in the middle troposphere (10–12 km), where SOPs are typically observed, the observational
values are also shown at similar vertical resolution for comparison. The average ozone mixing ratios
along with the corresponding standard deviations for the six events are compared between model
and observations in Table 1 for lower, middle and upper tropospheric altitudes.





The EMAC model is found capable of reproducing the altitudinal placement of the SOPs over the
central Himalayas during all six events. For example, on 20th Apr and 9th May the model shows the
peak ozone mixing ratios at 10.5 km asl, in agreement with the ozonesonde profiles. On other events,
such as on 11th Feb, 10th Mar and 25th Oct, the altitude of SOP differs slightly (by 0.5-1 km) between
model and ozonesonde profiles, except on 7th Jun (by 2 km). The aforementioned discrepancies in
the altitude of SOPs occurrence might be related to the model vertical resolution.
In addition to the altitude of SOPs occurrence, EMAC also quantitatively captures the ozone
enhancements. The model bias in simulating peak ozone mixing ratios is found to be varying from
about -45 nmol mol$^{-1}$ (7th Jun) to +34 nmol mol$^{-1}$ (9th May). The biases are found to be within
the variability of 1 standard deviation in 10–12 km altitude (28–59 nmol mol$^{-1}$) as calculated from
ozonesonde observations during spring over this site (See Table 1 and Ojha et al. (2014)).
However, the model generally overestimates the ozone mixing ratios in the lower troposphere by
about 11–24 nmol mol$^{-1}$ (Table 1) and shows some limitation in capturing less pronounced SOPs,
typically observed outside the winter-spring seasons. The bias in the absolute ozone enhancement
(-45 nmol mol$^{-1}$) as well as in the altitudinal placement of the SOP (by 2 km) are higher on 7th
Jun. However, these events were identified visually (Ojha et al., 2014) and here we show all for
completeness. The SOP events will be selected based on specific criteria in order to calculate the
frequency of occurrence, as discussed in detail in Sect. 3.3.
Possible biases between model and observations could arise from a variety of sources, most im-
portantly, the time-evolution of the SOPs (Supplementary material - Figure 1). Therefore, the limited
number of ozone profile measurements could lead to a temporal difference in the state of SOP evo-
lution being compared between model and observation. We tried to minimize this effect by applying
a weighted average algorithm, as mentioned above.
Overall, because the model is able to reproduce the occurrences of SOPs, their altitudinal place-
ments and the ozone enhancements over the central Himalayas, we use EMAC simulations to in-
vestigate the underlying processes (Section 3.2), the frequency of occurrences (Section 3.3) and the
effects on tropospheric ozone budget (Section 3.4).

## 3.2 Origin of SOPs

In this section, we analyze the EMAC simulated meteorological and chemical fields in conjunction
with backward air trajectories to investigate the origin of SOPs over the central Himalayas. Fig. 3
shows the vertical profiles of potential vorticity (PV), a tracer of stratospheric intrusions, during the
SOP events observed over Nainital. PV vertical profiles during all SOPs show layers of high values
coinciding with the altitude of SOPs.
The enhanced PV layers are found to be weaker during June and October as compared to events
during late winter and spring. PV values are found to be between 3.1 PVU (20th Apr) to 4.7 PVU
(11th Feb) at the SOP altitudes for the events occurring in winter-spring. Even during the less pro-





nounced events of early-summer and autumn, the PV values at SOP altitude are 1.8–2.5 PVU. The
PV values at the SOP altitudes suggest that the air masses showing very high ozone levels (SOPs)
are of stratospheric origin.

To quantify the amount of ozone transported from the stratosphere during the SOPs, we compare

the EMAC simulated vertical profiles of $O_3$ with $O_3s$ (Fig. 4). The model shows that nearly all
excess ozone at the SOP altitudes is of stratospheric origin, during the winter and spring, despite
of the fact that the SOPs are below the LRT, except on 10[th] Mar. Since the LRT over this region
is located significantly higher (Fig. 4, also see Naja et al. (2016)) than the altitude of SOPs, and
that the ozone in SOPs is found to be of stratospheric origin, we conclude that stratospheric air
masses are sandwiched between tropospheric layers at 10–11 km altitude. This result complements
previous studies primarily showing the altitudinal placement of SOPs at about 14 km near the Lower
Stratosphere (UTLS) (e.g. Reid and Vaughan, 1991; Hwang et al., 2007).

The contribution of tropospheric photochemical sources to the SOPs can be represented by the

difference $O_3$-$O_3s$, which is found to be large (about 50 nmol mol$^{-1}$), near SOP altitude on 7[th] Jun.
This could be a combined effect of deep convective mixing towards the onset of the summer monsoon
and weak horizontal winds (Ojha et al., 2014; Naja et al., 2016) leading to the accumulation of the
photochemically processed air masses of tropospheric origin.

In order to investigate the underlying dynamics that transport the stratospheric air masses, leading

to the SOPs over the Himalayas, we analyzed the backward air trajectories (Fig. 5), initialized over
Nainital at 10, 11 and 12 km, which are the typical altitudes of the SOPs (Fig. 2). The air mass trajec-
tories indicate rapid transport from the west, for example on 11[th] Feb, taking only two days for the air
masses to be transported across Africa and Middle-East and reach the Himalayas (Fig. 5). Further,
the locations of the tropopause folds occurred during the period of air trajectories are also shown.
The tropopause folds are mostly found in a belt between about 20 and 35°N, in agreement with
previous studies (Ŝkerlak et al., 2015). The air masses have been encountering extensive tropopause
dynamics along the path of transport, before reaching the Himalayas.

EMAC simulated $O_3$ vertical profiles along with the 5-day backward air trajectories are shown in

Fig. 6. The pressure variations of the air masses and tropopause along the trajectory are also shown.
In agreement with the analysis of PV and $O_3s$, there is no apparent downward transport of airmass,
as typically observed in altitude variations of the backward trajectories during many STT events (e.g.
Ma et al., 2014; Sarangi et al., 2014). Strong variability in the altitude of the LRT along the path of
the transport is seen, except for the event of 7[th] Jun. This variability in LRT appears to be associated
with the tropopause folds as shown in Fig. 5. Several shallow tropopause folds are seen along the
transport path, while deeper folds (medium) are only seen during 11[th] Feb and 9[th] May (also see
Ŝkerlak et al. (2015)). Intrusion of a significant amount of $O_3$ due to tropopause folds over the
Eastern Mediterranean and the Middle-East was shown by Akritidis et al. (2016). The combination
of very strong winds associated with the subtropical jets (Fig. 5, (Ojha et al., 2014; Naja et al.,





2016)) and this intense tropopause dynamics, enriching the troposphere with stratospheric ozone,
leads to the formation of SOPs over the Himalayas. Transport of stratospheric air during the analysis
period is not found on 7[th] Jun (Fig. 6), which explains the smaller ozone enhancement in this event,
probably related to the presence of some residual influences from previous days (Fig. 5 and 6).
The transport of ozone rich air masses from the stratosphere towards the Himalayas can be seen
more clearly in the longitude-pressure cross sections at 30°N (Fig. 7), and latitude-pressure cross
sections at 80°E (Fig. 8) for all the events and the day before. Fig. 7 reveals three geographical
regions viz. Northern Africa and Atlantic Ocean, Middle-East and northern South Asia, where the
intrusions of stratospheric air masses can be identified. Blobs of air masses characterized by high
PV values ( > 2 PVU) are also seen. Additionally, Fig. 8 shows a strong disparity in the stratospheric
influences at 80°E, with effects of STT mostly confined at latitudes higher than 25°N, and minimal
over the Southern India. This result based on EMAC simulations is found to be in agreement with
the study by Ganguly and Tzanis (2011) based on ozonesonde observations at three Indian stations.
**3.3   Frequency of SOPs**
The frequency of SOP occurrences was not estimated over Nainital from observations, due to the
availability of only 3-4 profiles in each month, however a tendency of higher frequency during spring
was noticed (3 events), as compared to other seasons (1 event per season) (Ojha et al., 2014). In this
section, we use long-term EMAC simulations, conducted for a period of 15 years (2000–2014), to
investigate the frequency of SOP occurrence and seasonality over the central Himalayas. Due to the
variability in the SOP altitude as well as the absolute enhancements during the SOPs, general / unique
criteria can not be defined. Therefore, we first select the ozone profiles in which Average Ozone
Mixing Ratios (AOMR) at 10–12 km, a typical altitude of SOP occurrence, are significantly higher
(at least by 50%) compared to average ozone in the lower troposphere. Additionally, to explicitly
select only the profiles which are SOPs (and not a direct intrusion over the Himalayas) the additional
criterion was applied that directly above the SOP the ozone mixing ratios are again lower (at least by
20%), so that selected profiles have a shape typical of SOPs, as shown in Fig. 2. These two conditions
can be mathematically expressed as
$AOMR_{10-12km} \geqslant 1.5 \times AOMR_{0-6km}$
and
$AOMR_{12-14km} \leqslant 0.8 \times AOMR_{10-12km}$
Further, the factors 1.5 and 0.8 representing an enhancement by 50% and reduction by 20% were
suitably varied, which confirmed the generality of the result (not shown). We calculated the fre-
quencies of occurrence in percentage for each month during 2000–2014, and converted these to an
average climatological seasonal cycle, with the year-to-year variation shown as standard deviation
(1-sigma) (Fig. 9).





The highest frequency of SOPs over the central Himalayas is found during the pre-monsoon sea-
son (MAM), followed by winter (DJF). The frequency of SOP occurrences over Nainital increases
steadily from January (2.7%) to May (10.8%), and abruptly declines in June (1.2%). The model does
not predict any SOPs during July–October. It should be noted that here we included only those events
as SOPs, which show enhancements by at least 50%, therefore some events with smaller enhance-
ments could be present during July–October. It is suggested that the more frequent stratospheric
intrusions during spring, combined with the stronger horizontal advection, lead to more frequent
SOP events. The effects of stronger cross-tropopause exchange and influx of the stratospheric air
masses during spring and winter over the Himalayas and surrounding regions, such as southern
parts of the Tibetan Plateau, have also been shown by Ŝkerlak et al. (2014, 2015). The frequency
of SOP events over this region is minimum during the summer monsoon season, as the weak hori-
zontal winds (Ojha et al., 2014; Naja et al., 2016) do not transport the ozone from STTs over large
distances. The frequency of stratospheric intrusions and tropopause folds over the Himalayas and
surrounding regions are lower during the summer monsoon (Cristofanelli et al., 2010; Putero et al.,
2016). Multiple tropopauses that can occur in winter and spring over the Tibetan Plateau are shown
to be absent during the summer monsoon season (Chen et al., 2011). Additionally, stronger vertical
mixing due to monsoonal convection inhibits high ozone layers to form and sustain. These findings
are in agreement with the ground-based ozone measurements in the southern Himalaya, where about
78% of the stratospheric influences were attributed to the PV structures and subtropical jet streams,
while monsoon depressions only account for 3% of the events (Bracci et al., 2012). Further, the
seasonality of SOP frequency derived from EMAC simulations is consistent with the conclusions
based on the limited number of observational profiles in Ojha et al. (2014). Next we determine the
enhancements in tropospheric ozone columns due to presence of SOPs over the central Himalayas.
**3.4   Effect of SOPs on Tropospheric Column Ozone**
Fig. 10 shows the climatological mean seasonal cycle of the Tropospheric Column Ozone (TCO)
in Dobson Units (DU) over Nainital from EMAC simulations over the period 2000–2014. TCO
values are calculated by integrating ozone mixing ratios up to the LRT, determined using the WMO
definition. To investigate the effect of SOPs on TCO, we compare three TCO values: First using
EMAC simulated $O_3$ values from all time steps, second by selecting only the time steps when there
is an SOP event as per the criteria discussed in Sec. 3.3, and third by taking all time steps when SOPs
do not occur.
TCO values for All-times and No-SOPs are found to be very similar, mainly due to the large
number of data counts (more than 1000 data counts in individual month), as compared to those in
SOPs (0-120 data counts in individual month). The maxima of TCO during May and June (54.7±5.9
and 55.0±4.4 DU respectively) are attributed to the intense solar radiation and high pollution loading
over northern India. While photochemical production of ozone is less efficient during the winter



(TCO: 33.7±3.6 to 37.6±5.8 DU) and the summer monsoon (e.g. 44.9±4.9 DU in August). Overall,
the EMAC simulated TCO seasonality from all data is found to be consistent with satellite data (Ojha
et al., 2012) over this region.
The occurrences of SOPs are seen to clearly enhance the TCO values during the winter, pre-
monsoon and early summer. The maximum enhancement in climatological average TCO value due
to SOPs is found during January, when TCO values during SOPs (43.5±4.6 DU) are higher by as
much as 9 DU (26%) compared to the non-SOP time steps (34.5±4.6 DU). The enhancements in
tropospheric ozone loading over the central Himalayas due to SOPs are estimated to be 4–9 DU
(7–26%) during January to June.
**4   Conclusions**
In this study, we used the EMAC model to investigate the layers of high ozone mixing ratios (SOPs)
in the middle-upper troposphere, observed over the central Himalayas in northern India. EMAC
successfully reproduces the occurrence, altitudinal placement and the relative ozone enhancements
during SOP events observed in ozonesonde profiles. The vertical profiles calculated by EMAC show
layers of high PV (1.8–4.7 PVU) coinciding with the altitude of SOPs suggesting the influences
from stratospheric intrusions. The analysis of $O_3s$ further shows that generally all excess ozone at
SOP altitudes over the Himalayas is transported from the stratosphere. Photochemically produced
(tropospheric) ozone is found to be significant only towards the onset of the summer monsoon.
Analysis of backward air trajectories in conjunction with EMAC simulated $O_3$ distributions and
tropopause dynamics revealed that stratospheric air masses are sandwiched between tropospheric
layers at 10–11 km altitude due to tropopause folds which are rapidly transported along the sub-
tropical jet to cause SOP structures over the Himalayas. Regions as far as northern Africa and the
Atlantic Ocean, the Middle-East and northern South Asia are found to be regions of stratospheric
intrusions that act as sources of high-ozone mixing ratios. The distribution of $O_3s$ showed that STT
effects have been confined at latitudes higher than about 25°N and are minimal over the southern
India.
We used long-term model simulations (2000–2014) to calculate the frequency of SOP occurrence
showing maxima during spring (about 11% of the time in May), while no SOPs were predicted
during the July–October months. This is consistent with results based on ozone soundings over the
region. The high frequency of SOPs during spring is attributed to the occurrence of stratospheric in-
trusions combined with rapid horizontal transport. The minima in the frequency of SOPs during the
summer monsoon are partially due to much weaker horizontal transport supplemented with stronger
monsoonal convective mixing. Model simulations were further used to investigate the effect of SOPs
on the TCO. The EMAC simulated TCO seasonality is in agreement with satellite data. SOP occur-
rence is found to significantly enhance the TCO over the region by 4–9 DU (7–26%).



*Acknowledgements.* The model simulations have been performed at the German Climate Computing Centre
(DKRZ) through support from the Bundesministerium für Bildung und Forschung (BMBF). DKRZ and its
scientific steering committee are gratefully acknowledged for providing the HPC and data archiving resources
for this consortial project ESCiMo (Earth System Chemistry integrated Modelling).



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

**Table 1.** A comparison of average ozone mixing ratios between ozonesondes and EMAC model for the lower, middle and upper troposphere during the six SOP events over Nainital

| Date | 2–7 km | | 7–12 km | | 12–17 km | |
|------|--------|--------|---------|---------|----------|----------|
|      | Sonde | EMAC | Sonde | EMAC | Sonde | EMAC |
| 20110211 | 50.7±4.1 | 69.5±3.6 | 85.8±52.1 | 112.3±52.3 | 135.9±22.9 | 173.7±42.6 |
| 20110310 | 67.6±8.1 | 86.8±1.8 | 120.5±52.0 | 134.8±39.0 | 131.9±57.4 | 184.7±76.9 |
| 20110420 | 85.8±7.7 | 96.9±7.0 | 147.1±37.3 | 136.8±28.6 | 151.4±41.6 | 151.0±36.7 |
| 20110509 | 83.0±13.3 | 101.6±5.3 | 104.8±34.7 | 154.8±25.3 | 132.9±15.4 | 140.7±17.7 |
| 20110607 | 83.1±7.6 | 107.0±6.1 | 132.4±30.2 | 110.2±10.0 | 119.8±21.6 | 111.9±35.2 |
| 20111025 | 57.1±3.3 | 72.3±1.7 | 84.4±26.8 | 86.7±17.2 | 123.3±37.6 | 130.4±31.7 |





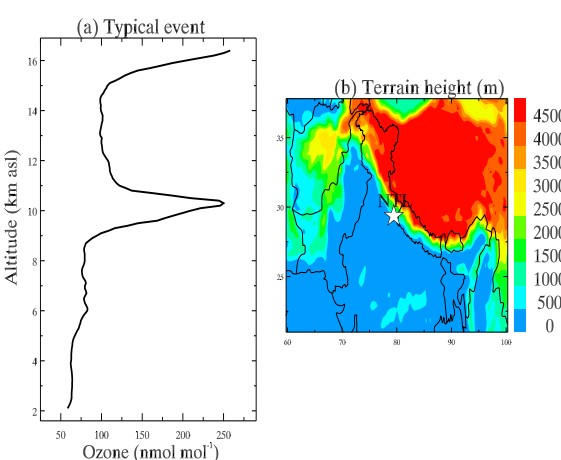

**Figure 1.** (a) A typical Secondary Ozone Peak (SOP) in an ozonesonde profile measured at 10–11 km altitude on 10[th] March 2011 over Nainital (Ojha et al., 2014). (b) Location of Nainital site in the central Himalayas shown in the topography map of the northern Indian region.



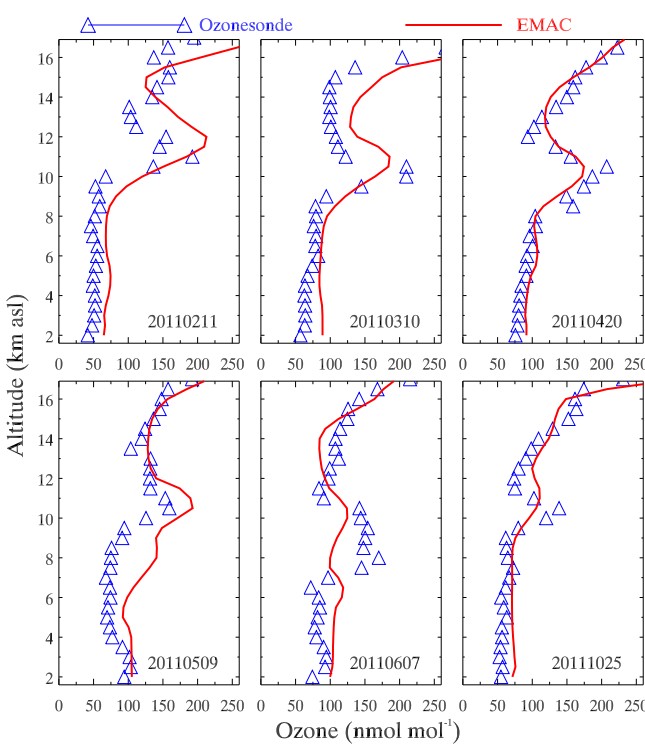

**Figure 2.** Comparison of EMAC simulated ozone profiles during the days of SOP events with ozonesonde observations over Nainital.





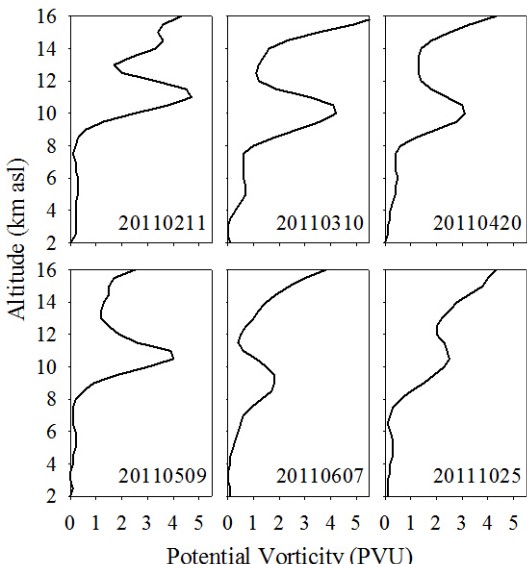

**Figure 3.** Vertical profiles of potential vorticity (PV) from EMAC simulations during the SOPs over Nainital.

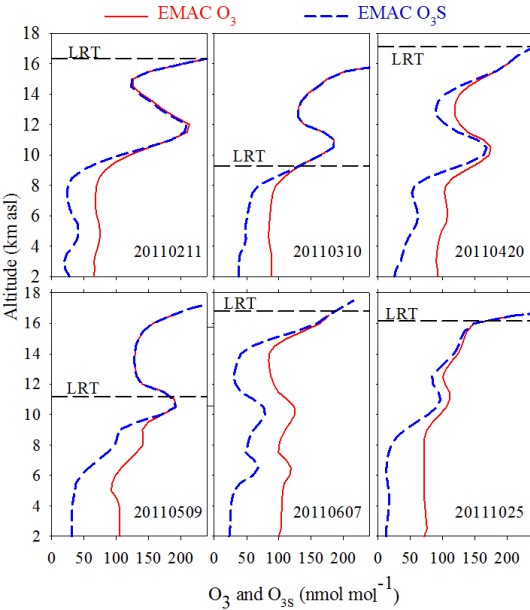

**Figure 4.** Vertical profiles of EMAC simulated ozone and stratospheric ozone tracer ($O_3s$) during the SOPs over Nainital. The height of the Lapse Rate Tropopause (LRT) from EMAC, calculated using the WMO definition, is also shown.





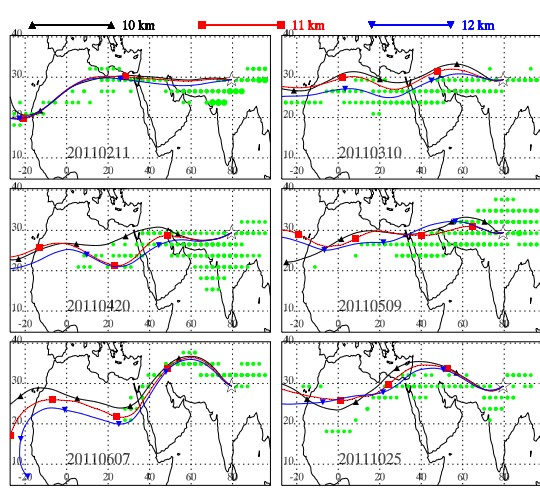

**Figure 5.** Backward air trajectories over Nainital for all events, with starting altitude of 10, 11 and 12 km. The difference between symbols on trajectories represent a time period of 1 day. The locations of tropopause folds 5 days prior to the event obtained from EMAC simulations are also shown. The location of Nainital site is shown by the star symbol. Small green circles represent shallow tropopause folds and bigger green circles (such as on 11[th]Feb) represent medium tropopause folds (see Sec. 3.2 for details).





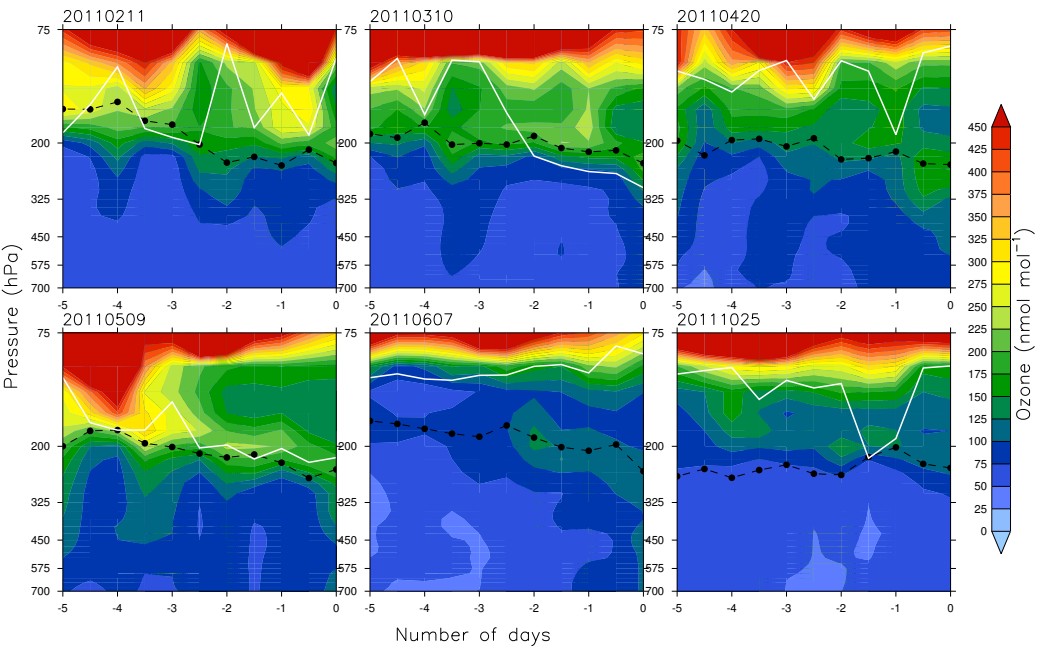

**Figure 6.** The vertical distribution of EMAC simulated $O_3$ along the trajectories with starting altitude of 11km over Nainital. The X axis shows the number of days back in time and the Y axis shows the pressure in hPa. The difference between two black circles here represents a time period of 12 h. The white line indicates the tropopause (LRT).





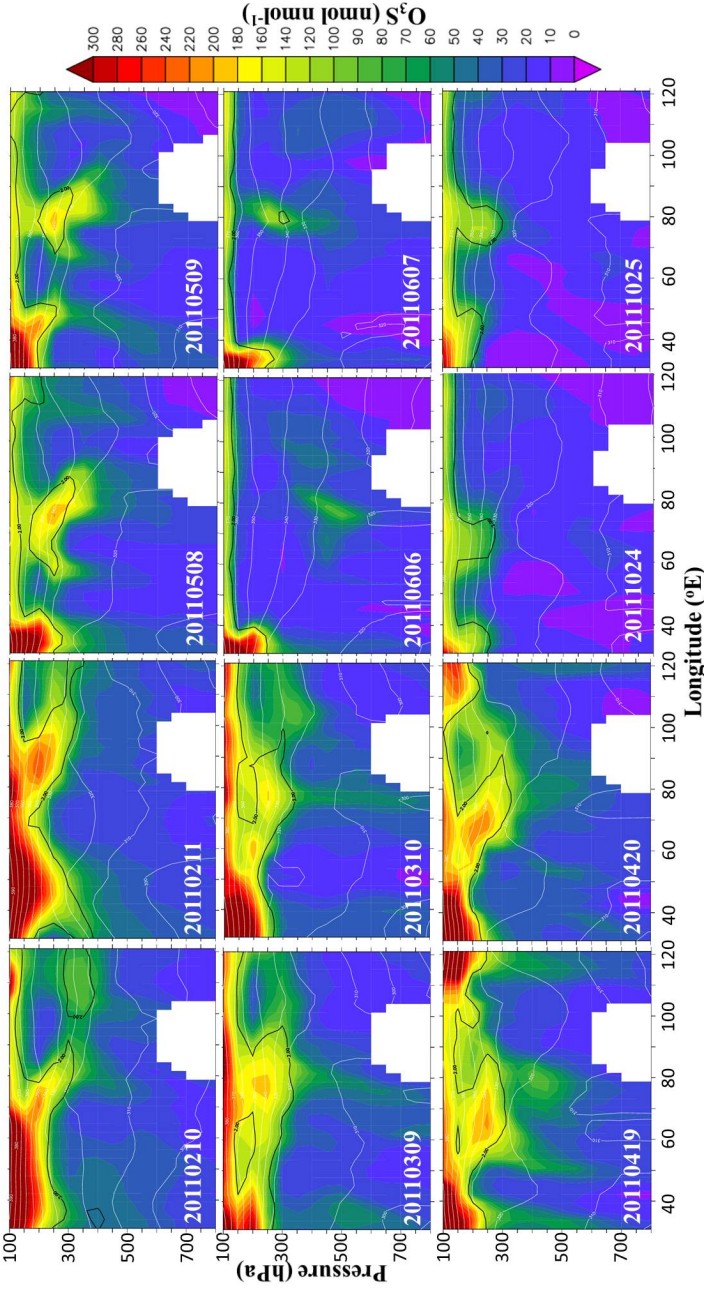

**Figure 7.** The longitude-pressure cross section of EMAC simulated $O_3s$ during all SOP days and a day before
the event. White lines denote the potential temperature (K) and the black line denotes the dynamical tropopause
at 2 PVU.




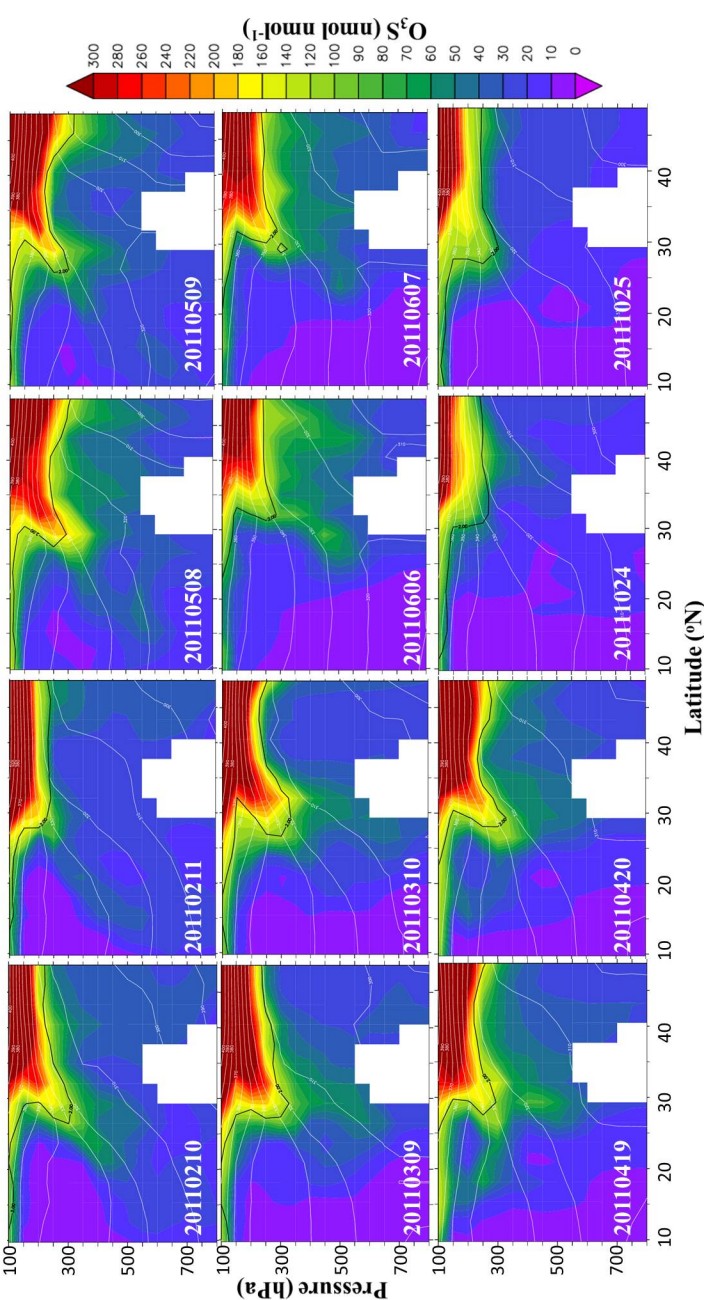

**Figure 8.** The latitude-pressure cross section of EMAC simulated $O_3s$ during all SOP days and a day before the event. White lines denote the potential temperature (K) and the black line denotes the dynamical tropopause at 2 PVU.




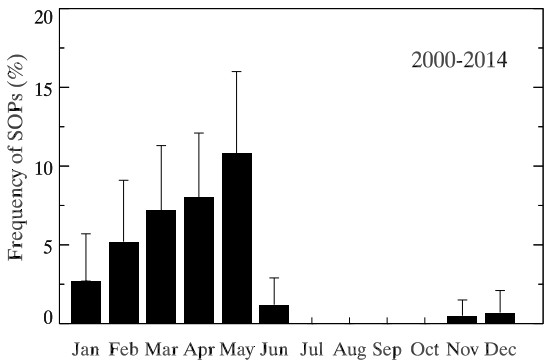

**Figure 9.** Annual cycle of SOPs occurrence frequency (%) over Nainital, calculated from the EMAC simulations for the period 2000–2014.

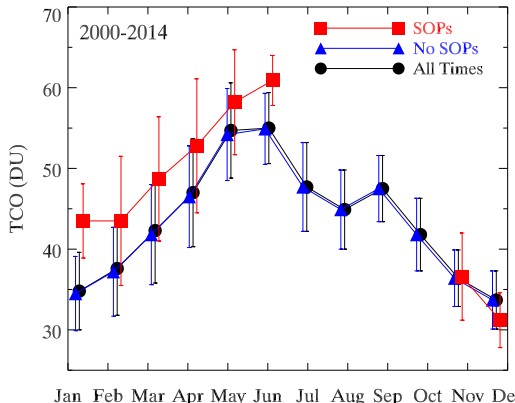

**Figure 10.** Annual cycle of EMAC simulated TCO over the central Himalayas calculated from 1) all EMAC time steps (All Times), 2) only the time steps having SOPs (SOPs), and 3) only when SOPs are not present (No SOPs) over the period 2000–2014.