# Peer review of "Secondary Ozone Peaks in the 2 troposphere over the Himalayas"

_Atmospheric Chemistry and Physics, 2016_

## Referee Comment (RC1) · Anonymous Referee #1 · 2 Dec 2016

GENERAL COMMENTS

This paper by Ojha et al. presents an interesting study about the occurrence of secondary ozone peaks (SOPs) in the troposphere over the Himalaya/Indian region.

The work is mainly based on the combined use of a limited set of vertical ozone sounding available at a single measurement site (Nainital in the Himalayan region) and on the outputs from the EMAC model, with the purposes of elucidating the processes leading to the occurrence of SOPs, characterizing their temporal variability and assessing their contribution to tropospheric ozone. The measurement data at Nainital were also used to evaluate the capacity of EMAC in reproducing SOPs.

The paper is well within the scopes of ACP and , potentially, it can seriously help in better clarifying this specific (but rather frequent) tropospheric phenomenon and its

implication on tropospheric O3 budget over the region.

However, I think that the authors should better take advantage of the long-term (2000 – 2014) ECAM data-set to provide a more robust characterization of this class of events both in terms of their origin, dynamical features and impact on long-term O3 variability.

Thus, I recommend publication, but after that some major efforts will be implemented towards this direction. I'm rather confident that the authors can implement the requested changes in a reasonable small amount of time.

SPECIFIC COMMENTS

In the introduction you should better describe the paramount importance of clarifying processes affecting tropospheric ozone variability in the Southern Asia and Himalayas, two global hot spot for climate, atmospheric composition and anthropogenic pressures (see e.g. http://www.unep.org/pdf/ABCSummaryFinal.pdf).

The verification of EMAC model capacity in reproducing SOP is based on a limited number of vertical soundings (only 6). The comparison provided by Figure 2 is encouraging about the ability of EMAC (despite the relatively coarse horizontal resolution 2.8 x 2.8 deg) in reproducing the SOPs. However, you should mention that this very limited amount of data prevent a systematic assessment. Did you try to inspect soundings at other locations in the same region (e.g. New Delhi, see http://woudc.org/data/explore.php) to make the data-set for verification larger? Can you provide some references of earlier works showing comparison of EMAC vertical ozone profiles with measurements (maybe Jöckel et al, 2016 can be profitably cited )?

Pag 6, line 169: I would like to see the bias expressed as %. This would better help in understanding the deviation of the model from the measurements.

Pag 6 line 176: "However, these. . .for completeness". I cannot understand this sentence. Do you mean that the selection by Ojha et al. (2014) is not accurate? Please, rephrase!

It is a pity that the "core" Section "3.2 Origin of SOPs" is discussing only the results from the six selected profiles at Naintal! I strongly encourage the authors to use the 15-year ECAM outputs to investigate in a more systematic way and for a long-term perspective this point. Also the back-trajectories investigation can be carried out for the whole 2000-2014 period by using NCEP re-analysis. I would suggest to use the SOP events identified over the period 2000 – 2014 and aggregate them on a seasonal basis to provide indication about the amount of ozone transported from the stratosphere during SOP (by comparing average $O_3$, $O_3s$ and PV vertical profiles).

Figure 5. Basing on the Figure caption, the TF locations 5 days before the events are reported in the maps. However, all the back-trajectories showed very fast transport: 5 days before the arrival to Naintal the air-masses were (at least) off of the north Africa western coast-lines. Thus, which is the relationship with the identified TFs? I suppose the authors would say that the TF DURING the air-mass transport were reported. . . Moreover, how long the back-trajectories are? No information are provided along the manuscript. . . Also seasonal composites over the period 2000 – 2014 about the spatial locations of tropopause folding related to SOP events can be presented (see e.g. Figure 4 by Putero et al., 2016 but for tropopause crossing). What about days without SOPs? I guess that no (or fewer) tropopause foldings were crossed by back-trajectories for these cases. . .To provide a "climatological" long-term perspective, you should also consider the possibility to present a composite for Fig. 7 and Fig. 8 as a function of the seasons for the period 2000 – 2014.

Pag 7, line 228: "This variability in LRT. . ..in Fig.5". It is not clear to me. Please, explain better this kind of association. . .

Pag 7, line 232: please define "medium".

Pag. 8, line 241: please provide in the text longitude boundaries for these regions.

Pag 8, line 243: despite your statement at pag 8 line 236, basing on that plot, it looks that a STE is actually occurring also for the June event (a tongue of air-mass rich in O3

extended down to 500 hPa southward than 30N)!

Section 3.4 The authors must provide some information about the long-term SOP trend over the region of interest: this information is very valuable also taking into account the current debate about the occurrence and attribution of tropospheric ozone trends (see e.g. http://www.igacproject.org/TOAR) . Trends in seasonal/yearly frequencies or physical features (e.g. altitude) of SOP and the related O3 contribution are detected? Also the information that no long-term trends were detected is nevertheless valuable. . .

Figure 10: I would add the percentage contributions of SOPs to monthly TCO values. What the error bars represent?

Conclusions In general this Section reports very important general statements about SOP but which are mostly based on the analysis of just 6 case studies (see lines 323-220). I would recommend to try to increase the robustness of these interesting hints by adopting a long-term perspective basing on EMAC simulation.

Line 335: "The minimum in the. . .mixing". I would also mention the northward displacement of subtropical jet stream during summer monsoon.

Line 339: are you able to provide any indication about the impact of this increase in terms of radiative forcing over the region?

TECHNICALS

Figure 1. I would skip the typical event plot since it is also reported in Figure 2.

Figure 6: x-axis and y-axis. I suppose the black line is the back-trajectory pressure level: it should be reported in the caption.

Figure 7 (Figure 8): please indicate in the caption the latitude (longitude) value for which the cross section is produced.

Figure 9-10: what the error bars represent?

---

## Referee Comment (RC2) · Anonymous Referee #2 · 13 Dec 2016

Review of Ohja et al., Secondary ozone peaks in the troposphere over the Himalayas

The authors use soundings from an Indian station (Nainital) sampled over the course of one year to identify 'secondary ozone peaks' (SOP). According to the authors 3-4 profiles are available per month. Six profiles are presented showing an SOP. A comparison with the EMAC model at T42L90 is used to extend the limited data set over a time period of 15 years (2000-2014) to assess the impact of such events on the ozone column over the Himalayan region. During the monsoon season they find virtually no SOPs over the region of interest. According to the authors such SOPs contribute 7-9DU of ozone to the tropospheric ozone columns during SOP occurrence. They also show, that the SOPs are only a minor effect and do not significantly enhance the ozone column over the whole year. The quantification of ozone transport across the tropopause is important and as such this study could in principle add to this.

[Figure]

Overall the paper is well written and the graphics are clear and appropriate. However, the paper needs some major clarifications: I missed clear descriptions of terms and definitions given the central topic SOP: How do the authors define an SOP? They only provide a definition for the model analysis later in the manuscript, but does this also work for the soundings, which have a much higher resolution? How do they distinguish an SOP from a tropopause fold or do they imply folds as SOPs? This is not clearly stated at all also in the introduction. Directly linked to this they don't discuss the transience or irreversibility of the phenomena, which are however crucial for the irreversibility of ozone flux and the persistence of the effect. I also missed a careful analysis of the transport and mixing process, as stated in the abstract. The authors should and could provide this, but currently they show coincident fields, but not a process. Given these points there I recommend the paper for publication after the following points have been addressed.

Major points: 1) In section 2 the authors should provide a clear definition for SOPs, which have been applied to the soundings. Further: What is the vertical resolution of the soundings and which role does the resolution of the sounding play for definition and the final column ozone estimate? The authors also do not discuss the effect of the limited vertical and horizontal resolution of the model. How many layers do they miss compared to high resolution sonde profile and how would this affect the number of peaks and the ozone column?

2) The authors should pay more attention to the reversibility of the SOPs. As long as the SOPs keep their high PV values as indicated in Figures 2-4, the ozone peaks will not permanently contribute to the tropospheric ozone budget, since they do not mix as shown in Fig.4 by the O3s. Figures 7-9 show O3s structures in the troposphere which are collocated to the tropopause (i.e. PV structure). The authors could e.g. diagnose the evolution of O3S on an isentropic surface relative to the evolution of PV to diagnose a persistent effect of the SOPs on tropospheric ozone. Maybe an additional plot of wind gradients or Richardson number would give some further indication for the process.

3) I suggest to calculate a statistical amount of trajectories in the model and evaluate the evolution of O3s, O3 and PV along the trajectories? I can't see, how the current Lagrangian analysis provides a robust view on any exchange on the basis of one trajectory per case and I suggest to remove Fig.5 and 6. At least the authors could show plots of ozone timeseries along the trajectories in Fig.6. Instead of the current Fig.6 the authors could plot the ratio of O3S/O3 to illustrate the stratospheric entry (with PV as contour to differentiate between transience versus irreversibility). This would much more strengthen the paper. Alternatively the authors could use the ERA Interim data, which drive the EMAC to perform trajectory calculations with a statistical amount of data. This would also much better help to identify the process of ozone transport and mixing into the troposphere by diagnosing PV change.

4) For the estimate of the effect of the SOPs on the tropospheric ozone column the authors should extend their analysis. As long as they don't account for the PV change, their results are not related to the tropospheric ozone budget. I suggest to compare in addition to Fig. 10 O3 and O3s for PV < 2 only for periods with and without SOPs. This would give the ozone which stays in the troposphere and leads to an enhancement during periods of SOPs, which would strengthen the importance of the results.

Minor comments: l.53: If SOPs occur in the lower stratosphere, how are these defined? They can't be the result of the same mechanism as tropospheric SOPs, are they comparable? l.100: Whats teh output frequency of the model?

l.117: "Tropopause folds are identified..." : How do the results compare to Sprenger et al,2003 or Skerlak, 2014 (over the Himalayas)?

l.146: Why don't you use a larger number of trajectories and perform a robust analysis?

l.155,156: Why is the model interpolated and not simply evaluated at the model levels, which would avoid interpolation errors particularly in the vertical? Is the output interpolated in time?

l.167-172: How do the relative ozone enhancements of compare to the observations instead of the absolute values?

l.285-287: 285-287: Clearify: What is meant with" PV structures and subtropical jet-streams"? Do you mean tropopause folds below the jet?
* * *

---

## Author Comment (AC1) · 25 Mar 2017

Please find the response to reviewer's comments and revised manuscript in the Supplement zip file .

Please also note the supplement to this comment:
http://www.atmos-chem-phys-discuss.net/acp-2016-908/acp-2016-908-AC1-supplement.zip

---

## Author Response (AR1)

**Response to Reviewers comments on "Secondary ozone peaks in the troposphere over the Himalayas"**

**Anonymous Reviewer #1**

**GENERAL COMMENTS:** This paper by Ojha et al. presents an interesting study about the occurrence of secondary ozone peaks (SOPs) in the troposphere over the Himalaya/Indian region. The work is mainly based on the combined use of a limited set of vertical ozone sounding available at a single measurement site (Nainital in the Himalayan region) and on the outputs from the EMAC model, with the purposes of elucidating the processes leading to the occurrence of SOPs, characterizing their temporal variability and assessing their contribution to tropospheric ozone. The measurement data at Nainital were also used to evaluate the capacity of EMAC in reproducing SOPs. The paper is well within the scopes of ACP and, potentially, it can seriously help in better clarifying this specific (but rather frequent) tropospheric phenomenon and its implication on tropospheric O3 budget over the region. However, I think that the authors should better take advantage of the long-term (2000 – 2014) EMAC data-set to provide a more robust characterization of this class of events both in terms of their origin, dynamical features and impact on long-term O3 variability. Thus, I recommend publication, but after that some major efforts will be implemented towards this direction. I'm rather confident that the authors can implement the requested changes in a reasonable small amount of time.

**Response: We thank the reviewer for the careful evaluation of the manuscript and his/her constructive comments and suggestions. The paper has been revised and now includes more analysis of long-term model simulations, as discussed in responses to the individual comments.**

SPECIFIC COMMENTS

**Comment 1:** In the introduction you should better describe the paramount importance of clarifying processes affecting tropospheric ozone variability in the Southern Asia and Himalayas, two global hot spot for climate, atmospheric composition and anthropogenic pressures (see e.g. http://www.unep.org/pdf/ABCSummaryFinal.pdf).

**Response: The suggested information is mentioned in the revised version (Page:3, Lines:86-95) as "Additionally, model simulations are required to both trace the source regions and quantify the effect of SOPs on the tropospheric ozone budget. Such investigations are of key importance as the Indo-Gangetic Plain (IGP) and Himalaya region are global hotspot regions in terms of anthropogenic pressures that could impose threats to Asia's water and food security (Ramanathan et al., ABC report, 2008). Satellite-based studies corroborate the high pollution loading over northern India and the nearby IGP including the Tropospheric Column Ozone (TCO) over South Asia (Fishman et al., 2003).The IGP is a regional hotspot of the so called "Atmospheric Brown Clouds (ABC)", consisting of brown haze formed by sub-micron size aerosol particles, emitted from a wide range of anthropogenic and natural sources. It has been shown that ABC reduce the amount of sunlight reaching the Earth's surface by as much as 10 to 15 %, and enhance atmospheric solar heating by as much as 50% (Ramanathan et al., 2007)."**

**Comment 2:** The verification of EMAC model capacity in reproducing SOP is based on a limited number of vertical soundings (only 6). The comparison provided by Figure 2 is encouraging about the ability of EMAC (despite the relatively coarse horizontal resolution 2.8 x 2.8 deg) in reproducing the SOPs. However, you should mention that this very limited amount of data prevent a systematic assessment. Did you try to inspect soundings at other locations in the same region (e.g. New Delhi, see http://woudc.org/data/explore.php) to make the data-set for verification larger? Can you provide some references of earlier works showing comparison of EMAC vertical ozone profiles with measurements (maybe Jöckel et al, 2016 can be profitably cited)?

**Response: The reference to Jöckel et al, 2016 is added for comparison of EMAC ozone fields with aircraft-based measurements from the IAGOS-CARIBIC program (Page: 7, Lines:206). It is shown that the simulation used in this work (RC1SD-base-10a) overestimates ozone concentrations, although mostly in the lower troposphere, while in the tropopause regions a reasonable agreement is obtained, compared to satellite and aircraft observations: "The seasonal cycle is reproduced, but the lower values in the troposphere are generally overestimated by up to 40% by the model. In the stratosphere, differences are smaller, as the model underestimates measurements by 5%, reaching 30% only in summer (Jöckel et al, 2016). Comparison with ozonesondes launched in Delhi has been added in the supplement (Fig. S2), which also shows an overestimation in the lower troposphere. Ozone mixing ratios in the middle troposphere show good agreement with the observations (Page: 7, Lines:204-205).**

**Comment 3:** Pag 6, line 169: I would like to see the bias expressed as %. This would better help in understanding the deviation of the model from the measurements.
**Response: Suggestion is incorporated.**

**Comment 4:** Pag 6 line 176: "However, these. . .for completeness". I cannot understand this sentence. Do you mean that the selection by Ojha et al. (2014) is not accurate? Please, rephrase!
**Response: No we only meant that the events were identified visually in that paper, while with availability of long-term data in this paper we make a criteria to calculate their frequency. The sentence is rephrased.**

**Comment 5:** It is a pity that the "core" Section "3.2 Origin of SOPs" is discussing only the results from the six selected profiles at Nainital! I strongly encourage the authors to use the 15-year EMAC outputs to investigate in a more systematic way and for a long-term perspective this point. Also the back-trajectories investigation can be carried out for the whole 2000-2014 period by using NCEP re-analysis. I would suggest to use the SOP events identified over the period 2000 – 2014 and aggregate them on a seasonal basis to provide indication about the amount of ozone transported from the stratosphere during SOP (by comparing average $O_3$ , $O_3s$ and PV vertical profiles).
**Response: Section 3.2 is revised to incorporate the reviewer's suggestions by analyzing 15-year EMAC outputs for a long-term perspective.**
**Average vertical profile of PV during SOPs, derived from a long-term model simulation (2000–2014), shows similar structure, as shown for the individual events. Average PV values during SOPs are found to be significantly higher (e. g. 3.0±1.3 PVU in winter, 1.8±0.5 during summer monsoon) as compared to timesteps without SOP (0.3±0.2 to 1.5±1.3) (Page: 7, Lines: 219-225 and Supplementary Fig. S3, Table S1).**

**Evolution of $O_3s$, $O_3$ and PV along statistical amount of trajectories is presented (New Fig. 6 in manuscript and Fig. S7, S8 in the supplement). Air masses are enriched the ozone of stratospheric origin during transport to Nainital causing SOPs. A significant fraction of trajectories during non SOP timesteps originates over the south west having lower $O_3s$ (< 90 nmol mol$^{-1}$). The trajectories which do get higher contributions of stratospheric ozone are found to be diluted during the transport making the enhancements above Nainital too small to be an SOP (Page:8, Lines 264-272).**

**SOP events identified over the period 2000 – 2014 are aggregated on a seasonal basis and average profiles of $O_3$ , $O_3s$ and PV vertical profiles are presented (New Fig. 5, Fig. S3 and Table S1). The amount of ozone transported from the stratosphere during SOPs is also indicated. The average amount of ozone transported from the stratosphere to the SOPs is estimated to be the highest during spring (162.5±40 nmol mol$^{-1}$), followed by winter (149.4 ± 35 nmol mol-1). In contrast the contribution of tropospheric photochemical sources to the SOPs is highest during the summer monsoon (30 nmol mol$^{-1}$) (Page-8, Lines:236-239).**

**Comment 6:** Figure 5. Basing on the Figure caption, the TF locations 5 days before the events are reported in the maps. However, all the back-trajectories showed very fast transport: 5 days before the arrival to Nainital the air-masses were (at least) off of the north Africa western coast-lines. Thus, which is the relationship with the identified TFs? I suppose the authors would say that the TF DURING the air-mass transport were reported. . . Moreover, how long the back-trajectories are? No information are provided along the manuscript. . . Also seasonal composites over the period 2000 – 2014 about the spatial locations of tropopause folding related to SOP events can be presented (see e.g. Figure 4 by Putero et al., 2016 but for tropopause crossing). What about days without SOPs? I guess that no (or fewer) tropopause foldings were crossed by back-trajectories for these cases. . .To provide a "climatological" long-term perspective, you should also consider the possibility to present a composite for Fig. 7 and Fig. 8 as a function of the seasons for the period 2000 – 2014.

**Response: Yes, we meant TFs DURING the air-mass transport, now clarified in the caption. This figure has been now moved to the supplement (Fig. S4) (Reviewer 2, comment 3). Length of trajectories (5 days) is now mentioned in the section 2.4 as well as in the figure caption. Seasonal composites of the spatial locations of folds (Fig. S6) shows higher frequency of occurrence during SOPs. The days without SOPs have minimal effects of $O_3$s transport due to fewer folds along the transport path, and dilution of any effects before reaching the Himalayas.**

**Comment 7:** Pag 7, line 228: "This variability in LRT. . ..in Fig.5". It is not clear to me. Please, explain better this kind of association. . .

**Response: We meant to say that the dramatic changes in tropopause pressure along the trajectory (e.g. from 100 to 200 hPa on 11$^{th}$ Feb) could be associated with the tropopause folds. The sentence is suitably revised.**

**Comment 8:** Pag 7, line 232: please define "medium".

**Response: The folds having a vertical extent of 200 to 350 hPa are defined as medium folds. This is mentioned in the revised manuscript. Further details are available by Škerlak et al. (2015).**

**Comment 9:** Pag. 8, line 241: please provide in the text longitude boundaries for these regions.
**Response: Suggestion is incorporated.**

**Comment 10:** Pag 8, line 243: despite your statement at pag 8 line 236, basing on that plot, it looks that a STE is actually occurring also for the June event (a tongue of air-mass rich in O3 extended down to 500 hPa southward than 30N)!

**Response: We agree. The statement that stratospheric effect is not found on 7$^{th}$ June is removed. As pointed out by the reviewer, text is also revised considering that some effects also reach southward than 30N (Abstract: Page 1, line 9; Results: Page 9, Lines:303-304; Conclusions: Page: 12, Lines 411-412).**

**Comment 11:** Section 3.4 The authors must provide some information about the long-term SOP trend over the region of interest: this information is very valuable also taking into account the current debate about the occurrence and attribution of tropospheric ozone trends (see e.g. http://www.igacproject.org/TOAR). Trends in seasonal/yearly frequencies or physical features (e.g. altitude) of SOP and the related O3 contribution are detected? Also the information that no long-term trends were detected is nevertheless valuable.

**Response: We evaluated trends in SOP frequencies on seasonal and yearly basis, however, as expected due to their origin from dynamical processes, frequency of SOPs discern strong**

**inter-annual variation as shown in the new Figure S9 in the Supplement. We discuss this in the revised manuscript (Page:13, Lines: 425-429) and provide relevant references.**

**Comment 12:** Figure 10: I would add the percentage contributions of SOPs to monthly TCO values. What the error bars represent?
**Response: The percentage contributions of SOPs to monthly TCO values is added. Error bars represent the standard deviation derived from the temporal variations over the period of 2000-2014. Now mentioned in the figure caption.**

**Comment 13:** Conclusions In general this Section reports very important general statements about SOP but which are mostly based on the analysis of just 6 case studies (see lines 323- 220). I would recommend to try to increase the robustness of these interesting hints by adopting a long-term perspective basing on EMAC simulation.
**Response: Conclusions are revised to add the additional results based on long-term model simulations (as mentioned in response to comment 5 also) (Page:12, Lines: 395, 399-401).**

**Comment 14:** Line 335: "The minimum in the. . .mixing". I would also mention the northward displacement of subtropical jet stream during summer monsoon.
**Response: Suggestion is incorporated.**

**Comment 15:** Line 339: are you able to provide any indication about the impact of this increase in terms of radiative forcing over the region?
**Response: Radiative forcing is not explicitly investigated in this paper. To provide an indication, a 4-9 DU higher tropospheric column ozone (due to enhancement at the SOP altitude) would correspond to an increase in surface temperature by 0.07 to 0.16 degree, as we take into account the vertical profile of ozone forcing (Lacis et al., 1990). This is discussed in the revised manuscript (Page: 13, Lines: 422-424).**

TECHNICALS
**Comment 1:** Figure 1. I would skip the typical event plot since it is also reported in Figure 2.
**Response: Suggestion is incorporated.**

**Comment 2:** Figure 6: x-axis and y-axis. I suppose the black line is the back-trajectory pressure level: it should be reported in the caption.
**Response: Suggestion is incorporated.**

**Comment 3:** Figure 7 (Figure 8): please indicate in the caption the latitude (longitude) value for which the cross section is produced.
**Response: Suggestion is incorporated.**

**Comment 4:** Figure 9-10: what the error bars represent?
**Response: The error bar in Fig. 9 represents the standard deviation of SOP frequency in each month among different years from 2000-2014 (Page: 10, Lines: 336-337). In Fig. 10 , it shows the standard deviation in the temporal variation of tropospheric ozone column among all the time steps during 2000-2014. This is now added in the figure caption.**

**Response to Reviewers comments on "Secondary ozone peaks in the troposphere over the Himalayas"**

**Anonymous Referee #2**

**GENERAL COMMENTS:** The authors use soundings from an Indian station (Nainital) sampled over the course of one year to identify 'secondary ozone peaks' (SOP). According to the authors 3-4 profiles are available per month. Six profiles are presented showing an SOP. A comparison with the EMAC model at T42L90 is used to extend the limited data set over a time period of 15 years (2000-2014) to assess the impact of such events on the ozone column over the Himalayan region. During the monsoon season they find virtually no SOPs over the region of interest. According to the authors such SOPs contribute 7-9 DU of ozone to the tropospheric ozone columns during SOP occurrence. They also show, that the SOPs are only a minor effect and do not significantly enhance the ozone column over the whole year. The quantification of ozone transport across the tropopause is important and as such this study could in principle add to this. Overall the paper is well written and the graphics are clear and appropriate. However, the paper needs some major clarifications: I missed clear descriptions of terms and definitions given the central topic SOP: How do the authors define an SOP? They only provide a definition for the model analysis later in the manuscript, but does this also work for the soundings, which have a much higher resolution? How do they distinguish an SOP from a tropopause fold or do they imply folds as SOPs? This is not clearly stated at all also in the introduction. Directly linked to this they don't discuss the transience or irreversibility of the phenomena, which are however crucial for the irreversibility of ozone flux and the persistence of the effect. I also missed a careful analysis of the transport and mixing process, as stated in the abstract. The authors should and could provide this, but currently they show coincident fields, but not a process. Given these points there I recommend the paper for publication after the following points have been addressed.

**Response: We thank the reviewer for the careful evaluation of the manuscript and his/her constructive comments and suggestions. The paper has been revised as discussed in responses to the individual comments.**

**Major points:**
**Comment 1:** In section 2 the authors should provide a clear definition for SOPs, which have been applied to the soundings. Further: What is the vertical resolution of the soundings and which role does the resolution of the sounding play for definition and the final column ozone estimate? The authors also do not discuss the effect of the limited vertical and horizontal resolution of the model. How many layers do they miss compared to high resolution sonde profile and how would this affect the number of peaks and the ozone column?
**Response: SOPs are basically a significant enhancement in ozone mixing ratios as compared to the lower troposphere (by at least 50%). Additionally these are not a direct downward transport and ozone levels above SOPs are again lower (here we considered at least 20%). The defining conditions for SOPs are added in the Section 2 (Page: 5, Lines:148-150).**

**Sounding data was reported originally at 100 m vertical resolution. As mentioned by the reviewer, the identification of SOP (and their effects) could be affected by the model vertical resolution, if this would be coarser than the vertical extent of SOPs (10-12 km: about 2km). As the EMAC simulations were conducted at a vertical resolution of 0.5 km (i.e. four times finer than the typical SOP extend) by using 90 vertical levels, we are able to reproduce all the six events. It must be stressed that also in the observations SOP were observed both in the high-resolution 100 m sounding data (Ojha et al., 2014) and in the 500-m resolution data (equivalent to model vertical resolution at the tropopause) used for the model evaluation (see Fig. 2).**

**Moreover, the EMAC modeling system (with the exact same horizontal and vertical resolution) was found capable to reproduce the observed (ERA-Interim) spatiotemporal features of tropopause fold occurrences (Akritidis et al., 2016), indicating that the current model resolution is sufficient for resolving similar processes near the tropopause region.**

**Comment 2:** The authors should pay more attention to the reversibility of the SOPs. As long as the SOPs keep their high PV values as indicated in Figures 2-4, the ozone peaks will not permanently contribute to the tropospheric ozone budget, since they do not mix as shown in Fig.4 by the O3s. Figures 7-9 show O3s structures in the troposphere which are collocated to the tropopause (i.e. PV structure). The authors could e.g. diagnose the evolution of O3S on an isentropic surface relative to the evolution of PV to diagnose a persistent effect of the SOPs on tropospheric ozone. Maybe an additional plot of wind gradients or Richardson number would give some further indication for the process.

**Response: The calculations of tropospheric ozone budget are revised by implementing the PV criteria suggested by the reviewer as described in the response to reviewer 2's major comment 4.**

**Moreover, to investigate the mixing of the transported stratospheric air with tropospheric air in the vicinity of SOPs, we present a turbulence-index (TI) (new Fig. 10), as described in Ellrod and Knapp (1992), to detect Clear Air Turbulence (CAT) areas and potential mixing, similar to the approach followed by Traub and Lelieveld (2003).**

**The enhanced TI values during the SOPs above Nainital indicate higher probability of mixing between stratospheric and tropospheric air, supporting the irreversible nature of the associated STT (Page: 9,10; Lines: 307-314).**

**Comment 3:** I suggest to calculate a statistical amount of trajectories in the model and to evaluate the evolution of O3s, O3 and PV along the trajectories? I can't see, how the current Lagrangian analysis provides a robust view on any exchange on the basis of one trajectory per case and I suggest to remove Fig.5 and 6. At least the authors could show plots of ozone timeseries along the trajectories in Fig.6. Instead of the current Fig.6 the authors could plot the ratio of O3S/O3 to illustrate the stratospheric entry (with PV as contour to differentiate between transience versus irreversibility). This would much more strengthen the paper. Alternatively the authors could use the ERA Interim data, which drive the EMAC to perform trajectory calculations with a statistical amount of data. This would also much better help to identify the process of ozone transport and mixing into the troposphere by diagnosing PV change.

**Response: Statistical amount of trajectories are computed and evolution of $O_3s$, $O_3$ and PV is presented (New Figures 6, S7, and S8). Air masses are enriched the ozone of stratospheric origin during transport to Nainital causing SOPs. A significant fraction of trajectories during non SOP timesteps originates over the south west having lower $O_3s$ (< 90 nmol mol$^{-1}$). The trajectories which do get higher contributions of stratospheric ozone are found to be diluted during the transport making the enhancements above Nainital too small to be an SOP (Page:8, Lines 264-272).**

**Further, as suggested in place of previous Fig. 6, we show the ratio $O_3s/O_3$ to illustrate the transport from stratosphere and its advection towards Nainital (Fig. 7 in revised version). This is discussed in the manuscript as "The $O_3s/O_3$ ratio is mostly found to be close to unity (≥0.9) near the altitude (pressure) of air mass trajectory during transport, except on 7th Jun and 25th Oct (0.5–0.8). The intrusions enriching tropospheric air masses with stratospheric $O_3$ are clearly visible. More specifically, a significant stratospheric contribution to tropospheric ozone is found in the upper/middle troposphere during the 5-day period before the event, with the associated PV values (< 2 PVU) indicating mixing of stratospheric air into the troposphere" (Page: 9; Lines: 277-282).**

**Comment 4:** For the estimate of the effect of the SOPs on the tropospheric ozone column the authors should extend their analysis. As long as they don't account for the PV change, their results are not related to the tropospheric ozone budget. I suggest to compare in addition to Fig. 10 O3 and O3s for PV < 2 only for periods with and without SOPs. This would give the ozone which stays in the troposphere and leads to an enhancement during periods of SOPs, which would strengthen the importance of the results.

**Response: Budget calculation is revised by accounting for PV change (Fig. 12 in the revised manuscript). The effect on tropospheric column ozone is found to be slightly lower (3.3-7.5DU; up to 21%) when PV criteria is applied, as compared to when PV criteria is relaxed and timesteps with PV higher than 2 are also included (4-9 DU; up to 26%) (Page: 12, Lines: 383-389).**

**Minor comments:** l.53: If SOPs occur in the lower stratosphere, how are these defined? They can't be the result of the same mechanism as tropospheric SOPs, are they comparable?

**Response: Here we focused on the SOPs in the troposphere. We find that stratosphere troposphere exchange is the main source of SOPs in the troposphere. Differential advection of ozone poor and ozone rich air could lead to secondary ozone peaks in the stratosphere (Lemoine, 2004).**

l.100: Whats the output frequency of the model?

**Response: 10 hours. Mentioned in the revised version (Page:4, Line: 112). Detailed description of the simulation can be found in Joeckel et al. (2016).**

l.117: "Tropopause folds are identified..." : How do the results compare to Sprenger et al,2003 or Skerlak, 2014 (over the Himalayas)?

**Response: The mean seasonal (DJF, MAM, JJA, SON) climatology of shallow ($50 \leq \Delta p < 200$ hPa), medium ($200 \leq \Delta p < 350$ hPa) and deep ($\Delta p \geq 350$ hPa) tropopause fold frequencies (%) over the period 2000-2014 are presented (Fig. S5), for intercomparison with the studies of Sprenger et al. (2003) and Škerlak et al. (2015).**

**The EMAC-simulated fold occurrences are generally in agreement with the findings of the aforementioned studies both spatially and temporally, especially for shallow folds which constitute the majority of folds. Moreover, in agreement with Škerlak et al. (2015), the fold maxima over the Himalayas are found during MAM and DJF, while the minimum fold frequencies are found during JJA.**

l.146: Why don't you use a larger number of trajectories and perform a robust analysis?

**Response: Trajectories at every time step are computed for May 2002 (the month having highest SOP frequency). Evolution of $O_3s$, $O_3$ and PV along the trajectories are analyzed for SOP and No SOP time steps (New Fig. 6, S7, S8). Also see response to your comment 3.**

**Comment:** l.155,156: Why is the model interpolated and not simply evaluated at the model levels, which would avoid interpolation errors particularly in the vertical? Is the output interpolated in time?

**Response: Interpolation errors are minimal as model's vertical resolution is also ~500 m on which we are taking the observational profiles for comparison. A time weighted mean of the model profiles have been obtained by weighing higher the profile which is closer in time of the observation (also see Ojha et al., 2016). We suggest (and verified) that this procedure would better include the temporal evolution, as compared to directly taking the profile closest in time.**

l.167-172: How do the relative ozone enhancements of compare to the observations instead of the absolute values?

**Response: The relative ozone enhancements are also reproduced, in general, with in the variabilities (see Table 1).**

l.285-287: 285-287: Clarify: What is meant with" PV structures and subtropical jetstreams"? Do you mean tropopause folds below the jet?

**Response: We have modified the phrase in the revised manuscript as follows:"PV structures, induced by fluctuations of the zonal flow and tropopause folds development along the subtropical jet-stream".**

[revised manuscript text omitted]